# Pulmonary valve tissue engineering strategies in large animal models

M. Uiterwijk[1], D. C. van der Valk[2,3], R. van Vliet[4], I. J. de Brouwer[4], C. R. Hooijmans[5,6], J. Kluin[1]*

1 Heart Center, Amsterdam University Medical Center, Amsterdam, The Netherlands, 2 Department of Biomedical Engineering, Eindhoven University of Technology, Eindhoven, The Netherlands, 3 Institute for Complex Molecular Systems, Eindhoven University of Technology, Eindhoven, The Netherlands, 4 Faculty of medicine, University of Amsterdam, Amsterdam, The Netherlands, 5 Department for Health Evidence Unit SYRCLE, Radboud University Medical Center, Nijmegen, The Netherlands, 6 Department of Anesthesiology, Radboud University Medical Center, Nijmegen, The Netherlands

* J.kluin@amsterdamumc.nl

**Data Availability Statement:** All relevant data are within the paper and its Supporting Information files.

## Abstract

In the last 25 years, numerous tissue engineered heart valve (TEHV) strategies have been studied in large animal models. To evaluate, qualify and summarize all available publications, we conducted a systematic review and meta-analysis. We identified 80 reports that studied TEHVs of synthetic or natural scaffolds in pulmonary position (n = 693 animals). We identified substantial heterogeneity in study designs, methods and outcomes. Most importantly, the quality assessment showed poor reporting in randomization and blinding strategies. Meta-analysis showed no differences in mortality and rate of valve regurgitation between different scaffolds or strategies. However, it revealed a higher transvalvular pressure gradient in synthetic scaffolds (11.6 mmHg; 95% CI, [7.31–15.89]) compared to natural scaffolds (4,67 mmHg; 95% CI, [3,94–5.39]; *p = 0.003*). These results should be interpreted with caution due to lack of a standardized control group, substantial study heterogeneity, and relatively low number of comparable studies in subgroup analyses. Based on this review, the most adequate scaffold model is still undefined. This review endorses that, to move the TEHV field forward and enable reliable comparisons, it is essential to define standardized methods and ways of reporting. This would greatly enhance the value of individual large animal studies.

## Introduction

Worldwide, biomedical engineers and physicians work in close collaboration to develop and improve tissue engineered heart valve (TEHV) prostheses. Their joined goal is to create a viable heart valve, overcoming the disadvantages of currently available heart valve prostheses such as limited durability [1–3], the need for anticoagulation [4] and the inability to grow with the patient [5]. Successful TEHVs should be dynamic structures, ultimately composed of specialized viable cells. In addition, TEHVs need an extracellular matrix (ECM) that can remodel in response to changes in local mechanical forces and maintain favorable strength, flexibility,

**Funding:** This work was funded by ZonMW, more knowledge with fewer animals (Projectnumber 114024134, 40-42600-98-433). C.R. Hooijmans is employee of SYRCLE. M. Uiterwijk is support from the Netherlands Cardiovascular Research Initiative (CVON 2012-01): The Dutch Heart Foundation, Dutch Federation of University Medical Centers, the Netherlands Organization for Health Research and Development and the Royal Netherlands Academy of Sciences. D.C. van der Valk is supported by the Gravitation Program "Materials Driven Regeneration", funded by the Netherlands Organization for Scientific Research (024.003.013).

**Competing interests:** The authors have declared that no competing interests exist.

**Abbreviations:** CCN-1, Cysteine-rich angiogenic protein 61; PCUU, Poly-Carbonate Urethene Urea; ECM, Extra cellular matrix; PDO, Poly(1,4-dioxan-2-one); EDTA, Ethylenediamine Tetra Acetic Acid; PG, Pressure gradients; HV, Heart valve; PGA, Polyglycolic Acid; NSVD, Non- structural valve deterioration; PHO, Polyhydroxyoctanoate; P(L,DL) LA, Poly(L-lactide-co-D,L-lactide); PV, Pulmonary valve; P3HB, Poly (3-hydroxybutyrate); SIS, Small intestinal submucosa; P3HB3HHx, Poly (3-hydroxybutyrate-co-3-hydroxyhexanoate); SVD, Structural valve deterioration; P4HB, Poly (4-hydroxybutyrate); TE, Tissue engineering; PCBU, Poly-Carbonate Bis-Urea; TEHV, Tissue engineered heart valve; PCL-UPy, Polycaprolactone- 2-ureido-4[1H]-pyrimidinone.

and durability; beginning at the instant of implantation and continuing indefinitely thereafter [6,7]. The basis of a TEHV is the scaffold. The scaffold provides a (temporarily) template and ideally functions as an instructive roadmap for cells to differentiate and support active tissue remodeling [8]. Scaffold materials are traditional categorized in synthetic (e.g., biodegradable polymers) or natural (e.g., animal donor) derived biomaterials. Each type has inherent benefits and challenges [9–12]. Over the last two decades, an extensive library of scaffold materials, different cell sources and cultivating processes have been explored and studied in large animal models. The first *in vivo* functional evaluation of a concept is often tested in the pulmonary valve position, because of the low-pressure circulation and easy access [13,14]. Subsequently, the high-pressure (aortic or mitral) position is tested, which is riskier and technically more challenging.

Synthetic scaffolds favor in terms of availability and control of fabrication. It is hypothesized that synthetic scaffolds must undergo full bio-resorption to create the patient's own cell-based heart valve and prevent a possible chronic immune response on scaffold remnants *in vivo* [15]. Thus, a synthetic scaffold should be biodegradable, in which the degradation of the scaffold synchronizes with the production of ECM in such way that the valve remains functional. Moreover, the scaffold biomechanics need to resemble native leaflets regarding stiffness [16,17] and flexibility. Natural scaffolds are decellularized valves or tissues derived from donor species. In contrast to the currently available bioprostheses, which are also derived from animal donors, natural scaffolds for TE purpose do not undergo a process of collagen crosslinking. Bioprostheses are chemically (e.g. glutaraldehyde) crosslinked to provide strength and diminish recipient rejection [18]. However, crosslinking results in non-viable tissues, in which cells are not able to migrate into the fixed matrix, making tissue renewal and growth impossible. In the non-crosslinked matrix of natural TE scaffolds, cells can migrate into the ECM and the matrix potentially retains natural components that provide cues for cell migration and differentiation, resulting in constructive remodeling [19]. Moreover, the natural donor derived scaffolds should potentially preserve their ECM architecture and consequently their biomechanical character. Still, it is a challenge to retain these properties after the decellularization process [20–22]. Both synthetic and natural scaffolds can be pre-seeded with cells or bioactive agents or can be cell free at time of implantation. Until now, it is not clear if pre-seeding of scaffolds contributes to the outcomes [23,24].

It is a challenge to get (and keep) an overview of all the combinations in the applied strategies, the advantages and disadvantages, and the results of TEHVs functionality. An overview of all publications would provide a helpful tool for scientists to fill-in gaps of knowledge, find a way through literature and enable proper comparison of their study outcomes with the appropriate studies and, most importantly, to find the scaffold with the most potential for our patients. Systematic reviews including meta-analysis of animal studies are less common than those of clinical studies, though not less important; they enable scrutiny of the validity of the preclinical evidence, they raise awareness of poor study design and ultimately encourage improvements in scientific rigor and reporting, and they provide transparency [25].

The objective of this review was to evaluate, qualify and summarize all available publications of TEHVs that were tested in the pulmonary position in large animal models. We performed a systematic review, an assessment of the quality of reporting essential items, and a meta-analysis on mortality and valve functionality.

## Methods

This systematic review was registered in an international database (PROSPERO register CRD42018092766) and reported according to the PRISMA guidelines [26]

## Search strategy and selection of articles

OVID Medline and the Embase databases were searched to identify all original articles concerning pulmonary valve scaffold implantations in large animals for tissue engineering purpose (Syntax see S1 Table). The final search was conducted on October 25th 2020. We used keywords for large animal models (porcine, ovine, canine, primates, caprine), pulmonary valve (PV) replacements and tissue engineering (TE), without time or language restriction. Results of the search were uploaded in the Early Review Organizing Software (EROS; Institute of Clinical Effectiveness and Health Policy, Buenos Aires, Argentina). EROS was used to randomly allocate the references of the database to two independent reviewers (MU, RvV, and/or IdB), who screened references for inclusion based on title and abstract according to the inclusion criteria. We excluded (systematic) reviews and editorials & conference abstracts. Natural derived cell free scaffolds that were chemically fixed (e.g., glutaraldehyde) prior to implantation were excluded. Full-text copies of all publications eligible for inclusion were subsequently assessed by two independent reviewers (MU, RvV, and/or IdB). In case of disagreements, the first author (MU) and a third reviewer (RvV or IdB) jointly decided whether exception was justified. Remaining articles eligible for full text reading were cross-checked for other relevant studies.

## Data extraction

Data was extracted independently by two reviewers (IdB, MU, and/or RvV). After data extraction, each reviewer verified the other reviewer's data entries and data entries were also verified by a third reviewer (MU, DvdV).

First, we extracted the following data on study characteristics: general characteristics of the study, animal Characteristics, follow-up time, surgical approach, anticoagulation treatment after surgery, and scaffold biomaterial. Second, we extracted data regarding TE strategies: decellularization (agent), sterilization method, pre-seeding (e.g., bioactive agent or cell source) prior to implantation and cell culturing location, and other non-foreseen experimental specific items. Finally, we extracted data on the outcome mortality/morbidity and valve function. Cause of mortality after valve intervention was categorized as structural valve deterioration (SVD), non-structural valve deterioration (NSVD) (e.g., operation related) or endocarditis [6]. If indicated, we extracted the timepoint of death.

## Quality of reporting assessment

Due to the nonrandomized and non (uniformly) controlled nature of most preclinical studies, no standard risk of bias analysis could be performed, as validated tools are unavailable for these types of studies. Instead, to identify risk of bias in the area of design and reporting for TEHV studies specific, we used a custom-made questionnaire [27] (S2 Table). The questionnaire includes five topics: animal characteristics, study design, adverse events, procedure and tissue engineering items. These five items were scored in 22 questions with 'yes' or 'no/unclear'. Subsequently, the reporting quality of a question was calculated as the number of studies scoring positive divided by the total number of studies. This was classified as good (> 75% of the studies), average (50–75%) or poor (< 50%). All available information per article was reviewed, including supplementary materials, references to previous work and appendices. Two investigators per study (MU, AV, IdB, and/or AD) independently assessed the reporting quality of the included references.

## Meta-analysis

Mortality and valve functionality, more specific pulmonary valve regurgitation (PR) and the mean transvalvular pressure gradient (mPG), were assessed in the meta-analysis. The effect

estimates of single groups were presented and pooled (because no control group data regarding native or sham operated animals was available).

To asses mortality, the number (n) of animals that died or were terminated before the planned date were extracted and reported as fraction (%) of the total number of animals of the allocated experimental group. For valve functionality, data was extracted as raw data or group averages in case standard deviation (SD) or standard error (SE) and number of animals per group (n) were reported or could be recalculated. If one article studied the effects of two or more scaffold variants, methods or follow-up time, these groups were analyzed as independent comparisons. In cases data could not be extracted from the text but was only presented graphically, we used a universal on-screen digitizer (Fiji; ImageJ version 2.0.0) to quantify the data. When several time points (repeated measurements) in one subject (animal) were investigated for valve function, the time point at end of follow-up was extracted. In case this last time point only contained a single animal (n = 1), the data from animals in the previous timepoint was used. In case all subgroups within one study consisted of only one animal (n = 1) and subgroups were sufficiently comparable, these animals were combined as if they comprised one group. Subsequently, statistical analyses were performed in Comprehensive Meta Analyses software (CMA version 2.0). Forest plots were used to display the mean effect sizes. Data are expressed as effect size (ES) with 95% confidence intervals. In case there were more than two independent experiments, the event rates or means were pooled using a random effects model which takes into account the precision of individual studies and the variation between studies and weights each study accordingly. In case the median and range was reported, the mean and standard deviation was calculated [28]. To determine the study heterogeneity $I^2$ was used. Subgroups were predefined according to scaffold material and cellular state at time of implantation (cellular or acellular). The results of subgroup analyses were only interpreted when subgroups contained at least data from 3 independent studies or 5 experiments per subgroup. For subgroup analyses, we adjusted our significance level according to the conservative Bonferroni method to account for multiple analyses (p* number of comparisons).

To assess the possibility of bias resulting from the time point of the echocardiographic measurement, the conducted follow-up time of each comparison (included in the meta-analyses) was plotted in a box-plot and visually evaluated on asymmetry by two reviewers (CH and MU).

Sensitivity analyses was conducted for the echocardiographic method (TTE, TEE, intracardiac and epicardial) and non-parametric or parametric reported data in the continues data.

## Results

### Literature search and screening

The database searches yielded 762 titles. After removing duplicates, 561 papers entered the title abstract screening phase. During title and abstract screening, 461 papers did not meet our predefined inclusion criteria, resulting in 100 papers for full text screening. Screening of the reference list of these papers did not result in any new references. Finally, 80 papers [3,15,29–107] were included in this review. The study selection process is illustrated in Fig 1.

### Study characteristics

The 80 included papers studied 693 animals that received a TEHV in pulmonary position. The papers were published between 1995 and 2020. We saw an increase of publications on synthetic scaffold implantations over these 25 years (Fig 2A). The animal strains used in the studies were: sheep (n = 557/693; 80%), pigs (38/693; 5%), primates (20/693; 3%), dogs (58/693; 8%) or goats (20/693; 3%). Animal sex was reported in 33 papers (41%), including 274 animals.

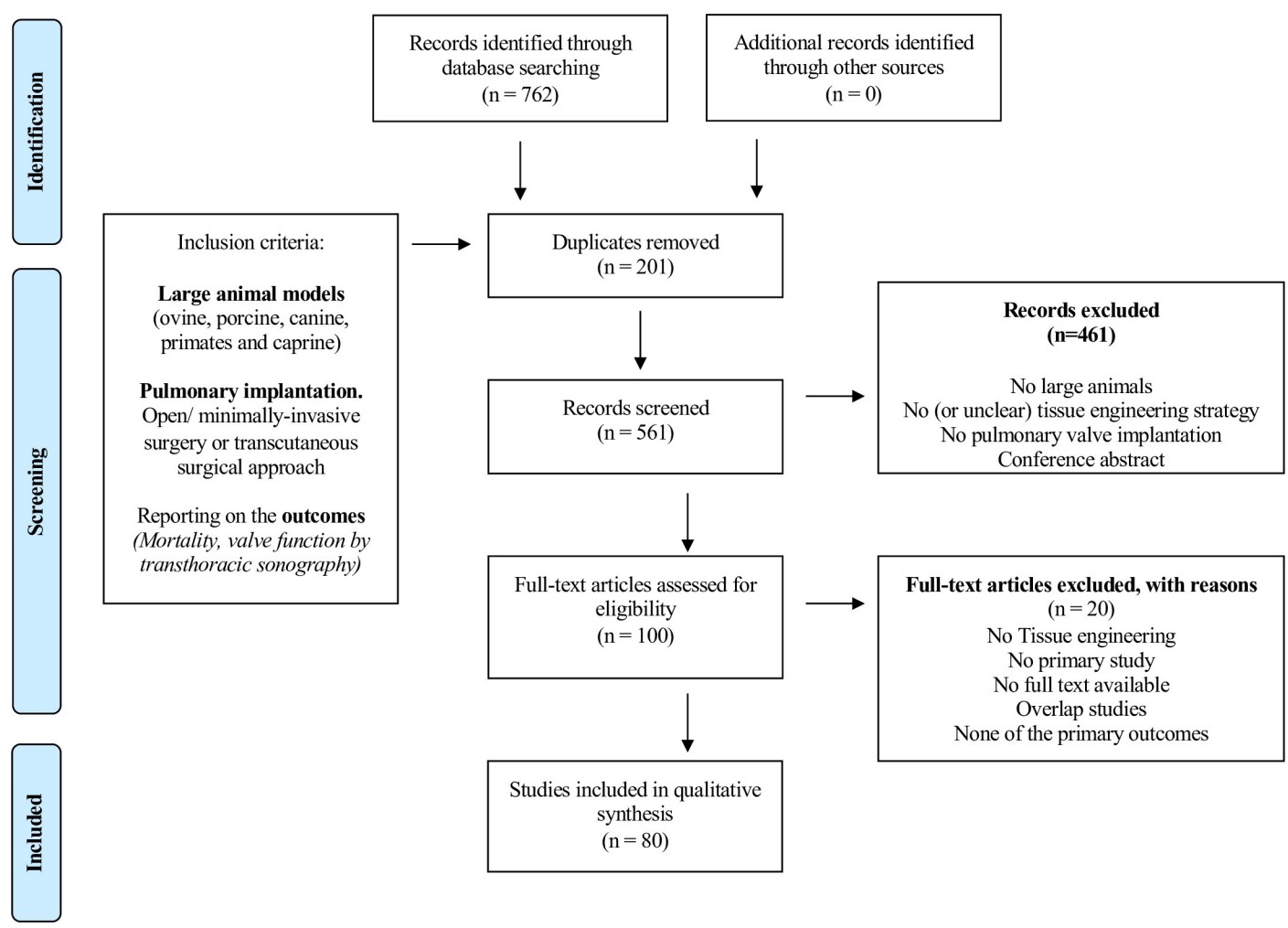

**Fig 1. Flowchart outlining the protocol adapted from PRISMA guidelines.**

If sex was reported, 93% (249/268) of animals were female and 7% (19/180) male. The follow-up time of the studies ranged from acute (hours) to 24 months. Studies of synthetic valves were evaluated after a mean follow-up of 3.4 months (modus 1 month, range 1 hour– 24.0 months). Natural derived scaffolds were evaluated after a mean follow-up of 5.0 months (modus 6 months, range 0.1–22.5 months). Postoperative use of anticoagulation therapy was described in 24 studies (30%), of which 13 studies (16%) explicitly mentioned not to use anticoagulation. Details of the anticoagulation treatment can be found in S3 Table. An overview of the characteristics of the included publications on synthetic and natural scaffolds can be found in the supplement (S4A and S4B Table) and results are illustrated in Fig 3B–3D.

## Scaffold characteristics and tissue engineering strategies

A variety of TEHV strategies was reported. Synthetic (*not created by/in nature*) and natural (*native tissue/donor derived)* scaffolds were used in 32% (223/693) respectively 68% (470/693) of animals. Of these two scaffold types at the start (synthetic and natural), we identified 11 different strategies to the moment of implantation. These strategies concealed (Figs 3A and 4)

**A. Publication on synthetic and natural scaffolds.**

**B. Number of animals used per year.**

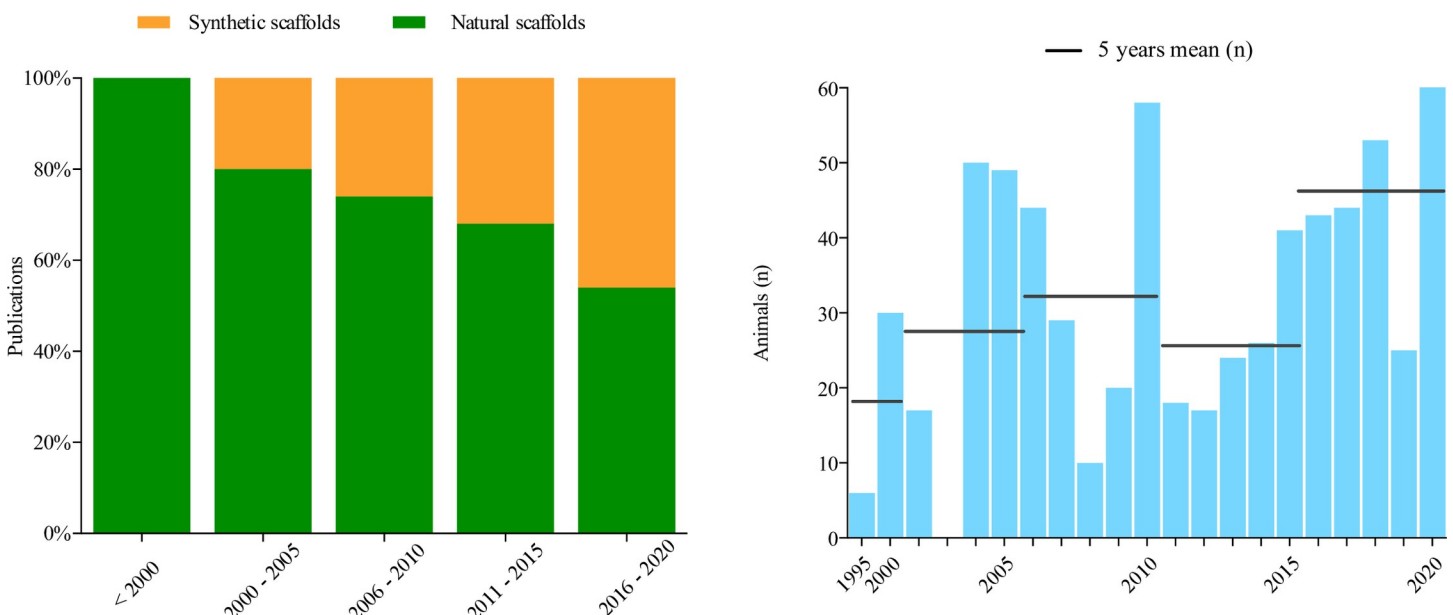

**Fig 2. A.** Percentage of publications of studies on TEHV scaffolds shifted from mainly natural-based scaffold designs before 2015, to synthetic scaffolds designs after 2015. **B**. Total number of animals used per year in studies on TE pulmonary valve implantations showed increase of (mean) numbers of animals over the last 20 years.

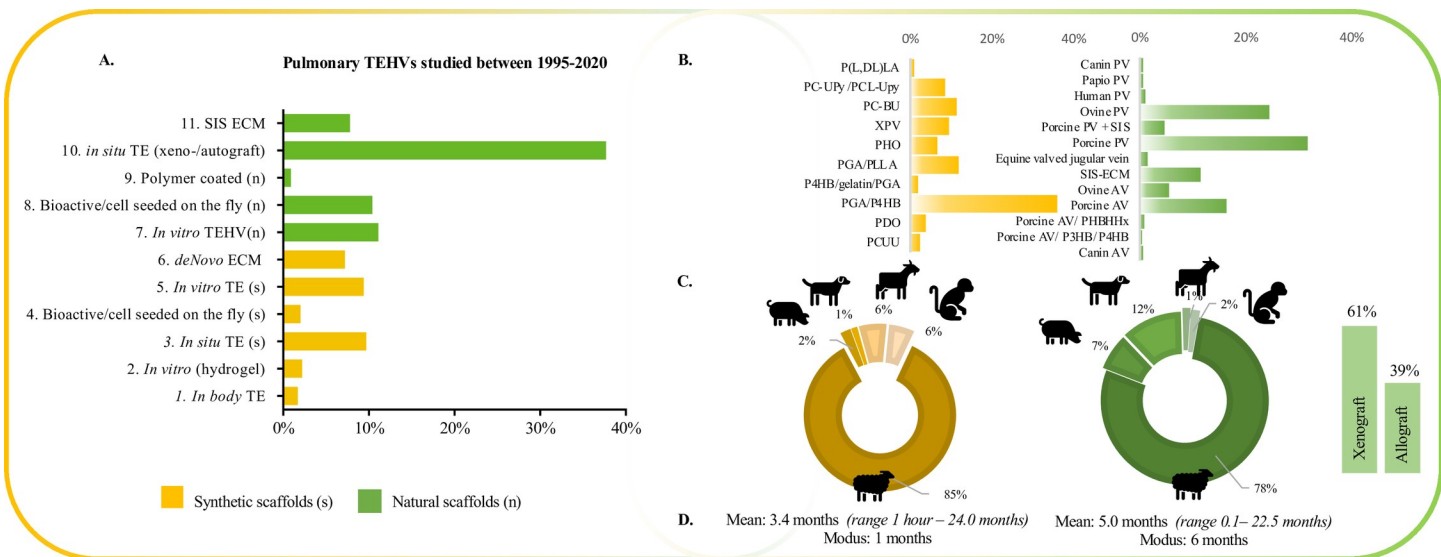

**Fig 3. Study characteristics in studies on synthetic- (yellow/orange) and natural (green) based scaffolds. A.** Mainly acellular *in situ* TE was performed with natural scaffold, also mentioned as decellularized xeno-/autografts. Histogram presents percentage of valves of total (n = 693) valve implants. **B.** Nine different polymers and one not specified (XPV) type were identified in studies analysing polymer- based synthetic scaffolds. Natural valves originated most often from pigs. **C.** The sheep was the most used animal model in studies on synthetic as well in natural scaffolds. Natural scaffolds were mostly implanted as xenograft **D.** The most used follow-up time for synthetic scaffolds was shorter (modus 1 month) compared to that of natural scaffolds (modus 6 months). *AV: Aorta valve. BU; Bis-urea ECM; Extracellular matrix P(L,DL)LA; Poly (L-lactide-co-D,L-lactide). P4HB; Poly-4-hydroxybutyrate PC; Poly-carbonate PCL; Polycaprolactone PCUU; Poly-carbonate urethene urea PDO; Poly(1,4-dioxan-2-one) PGA; Polyglycolic Acid. PHO; Polyhydroxyoctanoate PV; Pulmonary valve. SIS; Small intestinal submucosa TE; Tissue engineering TEHV; Tissue engineered heart valve. UPy; 2-ureido-4[1H]-pyrimidinone.*

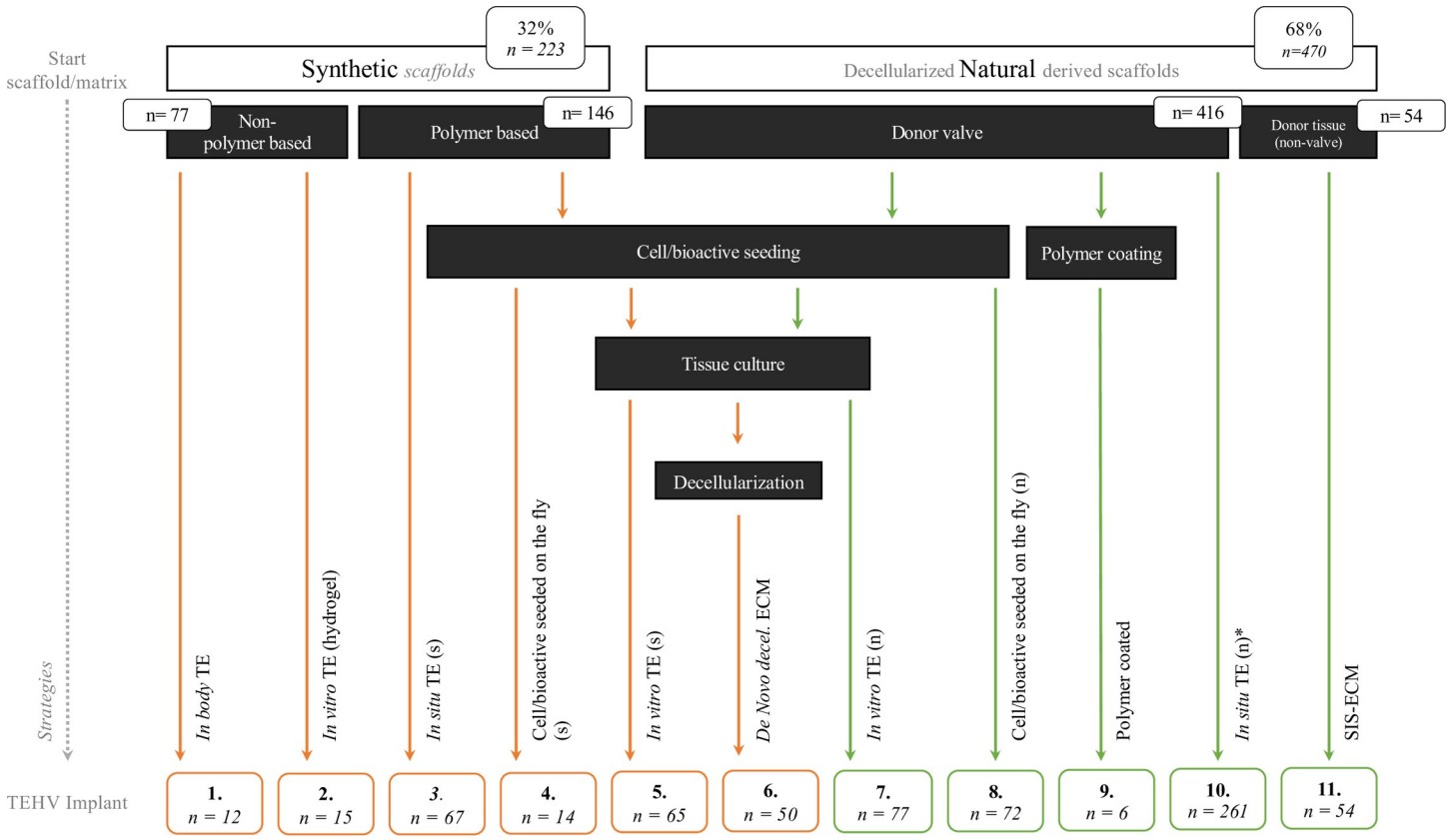

**Fig 4. Overview of strategies in tissue engineering of heart valves.** Start with the scaffold materials (synthetic or natural), subsequent the pathways of preparation techniques prior to implantation. Finally, 11 strategies were identified. *n = numbers of valves represented in each group. TE; Tissue engineering. SIS-ECM; Small intestinal submucosa extra cellular matrix.* *Decellularized. Xeno-/autografts.

pre-seeding, tissue culturing, coating and decellularization processes, in single or combined order.

The polymer-based synthetic scaffolds (196/223; 87.9%) were directly implanted (67/223; 30.0%; *in-situ* TE), pre-seeded and cultured (65/223; 29.1%; *in vitro TE*) or decellularized after cell culturing (50/223; 22.4%; *decellularized de novo*). Of the polymer-based synthetic scaffolds, 10 different polymers (blends) were used (Fig 3B). In addition, 12 valves were (hand-) made of connective tissue sheets which were cultured subcutaneously in dogs (12/223; 5.4%; *in body TE*). Another 15 valves (15/223; 6,7%) were made of tissue, engineered from cultured (ovine) fibroblast in hydrogel.

The natural scaffolds were mostly derived from donor valves (416/470; 88.5%;) and decellularized. Decellularization methods of the natural valves included: non-ionic detergents (e.g., Triton X-100), ionic detergents (SDS or sodium deoxycholate), zwitterionic detergents (hypotonic and hypertonic solutions, EDTA), or enzymatic (endonucleases) methods. The natural scaffold derived from donors were directly implanted (261/470; 55,5%; *in situ* TE), pre-seeded and cultured (77/470; 16,4%; *in vitro* TE), seeded on the fly with cells or bioactive agents (72/461; 15.3%;) without a culture period, or coated with a polymer (P4HB) (6/461; 1.3%). In addition, 54 (11.5%) valves were hand-made from commercially available porcine derived decellularized small intestine submucosa ECM sheets (SIS-ECM).

Sixty-one percent (287/470) of the natural valves were implanted as xenograft and 39% as allografts.

Most donor valves were harvested from pulmonary position (297/416; 71.4%) or from aortic position (112/416; 26.9%). One study (7/416; 1.7%) implanted (equine) jugular vein valves in the pulmonary position in sheep.

## Scaffold decellularization, seeding, culturing and coating

If scaffolds were pre-seeded, a variety of methods and cells was used. Mostly, seeding of the synthetic scaffolds (n = 22 studies) was conducted with vascular or bone marrow derived (BMD) (autologous) cells. These were mostly typed as (myo)fibroblast cells and in several cases combined with seeding of endothelial cells (ECs). Exceptions to mention here are neonatal human derived dermal fibroblast [38,101], human vascular derived fibroblasts [54], and BMD mononuclear cells [37,40,100] or a combination [34]. Pre-seeding of natural scaffolds (n = 14 studies) was also mainly performed with vascular derived (myo)fibroblast (MFB) like cells, sometimes combined with (progenitor) ECs and/or smooth muscle cells. Three studies used BMDs mononcluear cells [71], endothelial progenitor cells [57] or EC-like and MFB-like cells [61]. Three studies with naturel decellularized scaffolds used bioactive agents to stimulate specific cell adhesions, respectively fibronectin alone [3,58], or combined with stromal derived factor [60], fibronectin with or without hepatic growth factor (HGF) [68], or cysteine-rich angiogenic inducer 61 (CCN-1) [70]. Two studies evaluated naturel decellularized scaffolds coated with polymer PHBHHx 3–5% [55] or a blend of 82% P3HB and 18% P4HB [56]. One study conditioned the synthetic scaffold with transforming growth factor ß-1(TGF-ß1) [101]. If valves were cultured *in vitro*, this was done in static conditions in six studies using natural scaffolds [30,32–36] and in seven studies using synthetic scaffolds [3,57,61,62,67–69]. In some cases, the static condition was followed by a dynamic environment. In eleven studies on synthetic [29,32,34,49–54,101,105] and four studies on natural scaffolds [64–66,70], only dynamic (pulse and/or flow) systems were used to mature cells or the novo ECM in vitro (S5A and S5B Table).

## Reporting quality

The assessment on reporting quality is illustrated in Fig 5 and supplement S6 Table. Animal characteristics (Q1- Q6) were generally well reported, except for animal gender, which was poorly reported (41% of the studies). The items concerning the study design (Q7- Q11) were poorly reported. In 56% (45/80) of the studies, some sort of control group was present (Q9) to evaluate the echo results. Studies described the use of negative or positive controls (e.g., cellular versus acellular scaffolds) in 89% (40/45 studies), comparative controls (e.g., clinically used biological valves) in 9% (4/45), or a combination of these in 7% (3/45). In one study, a sham-control [81] was used. To get more insight in the reasons not to use a control group, we asked additional questions (Fig 5B). This showed that 9% (3/35) of authors mentioned in their manuscript the reason why they did not use a control group. The other studies (32/35; 91%;) did not specify the reason for absence of a control group. Of the latter, 31% (10/32) of the studies could be qualified as a pilot or feasibility study. In the other 69% (22/32), one scaffold variant was studied and evaluated at different time points. Random allocation (Q10) of animals to their experimental group (if applicable) was reported in five (11%) studies. Blinding of qualitative functional outcome assessment (Q11) was described (if applicable) in only 2 of the 72 (3%) studies.

The third and fourth topic concerned items regarding the adverse events (Q12-15) of the surgical procedure and composition (Q16-Q17) of the valves, which were almost all well reported (>75%), except the timepoint of the dropouts (Q14), that was not clearly reported in 67% (26/39) of the studies. The diameter of the implanted valve was described in 68% (54/

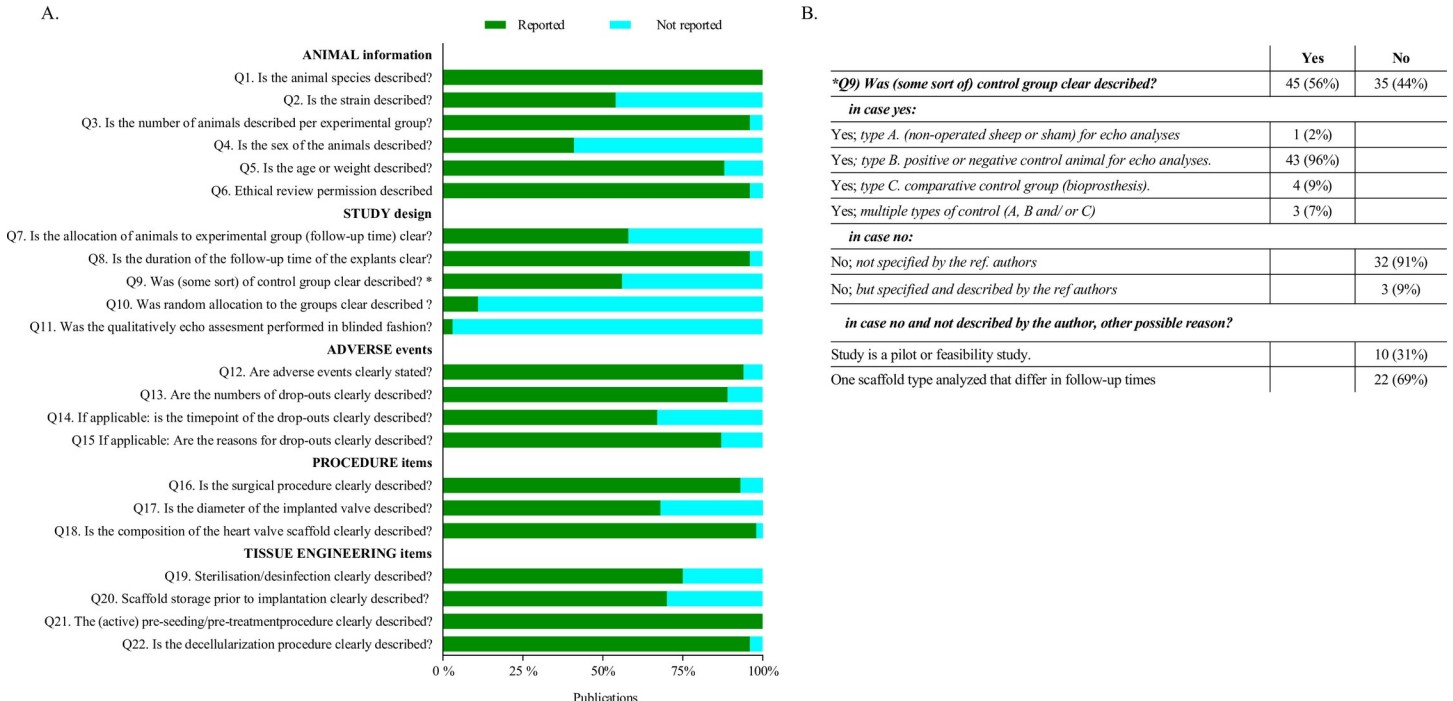

**Fig 5. Quality assessment of included articles (total n = 80). (A)** Quality of reporting on animal information, study design, adverse events, procedure items, and specific tissue engineering items showed a general moderate report of all items except the random allocation to the groups (selection bias) and blinding of the assessment of echo quality (detection bias). **(B)** Specification of the control variants showed a general preference for the use of a positive or negative control if a control was used. The absence of control groups consisted mainly of an unspecified description by the referenced authors, but these studies usually only showed results for one subgroup.

80) of studies. The last topic concerned the TE strategies (Q18 -Q22). The sterilization method (Q19) and banking prior to implantation (Q20) were moderately reported (75%; 59/79 and 30%; 22/73 respectively). Pre-seeding and decellularization procedures (Q21 and Q22 respectively) were well reported (38/38;100% and 54/56; 96%).

## Meta-analysis

**Mortality.** Data on mortality could be extracted from all studies on synthetic (27 comparisons, n = 196) and all studies on natural scaffolds (51 comparisons, n = 493). The overall pooled estimated mortality rate was 17% (CI 95% [13–20%]) and did not differ between synthetic (18% (CI 95% [12–27%]) and natural scaffolds (16%; CI 95% [12–20%], $I^2$ = 0). Subgroup analyses, applicable on 7 out of 11 TE strategies, did not show significant difference in mortality (Figs 6A and 7A and S7 Table).

**Valve regurgitation.** Twenty-one studies on synthetic and 32 on natural scaffolds, containing 35 and 75 independent comparisons (n = 122 and n = 251), assessed valve regurgitation by sonographic imaging. There was no statistically significant difference in fraction of moderate/severe regurgitation between synthetic and natural scaffolds nor in the subgroup analyses, applicable in five out of the 11 strategies (Figs 6B and 7B).

**Valve pressure gradient.** Fourteen studies on synthetic and 20 on natural scaffolds, containing 22 and 42 comparisons (n = 60 and n = 93), reported the mean valvular pressure gradient (non-invasively measured). Mean pressure gradient was significantly (*p = 0.03*) higher in the synthetic scaffolds than in the natural scaffolds, resp. 11.6 mmHg (95% CI [7.3–15.9]) versus 4.7 mmHg (95% CI [3.9–5.4]; $I^2$ = 100.0) (Fig 6B). Subgroup analyses were only conducted for the groups containing at least 5 comparisons and a minimum of three independent

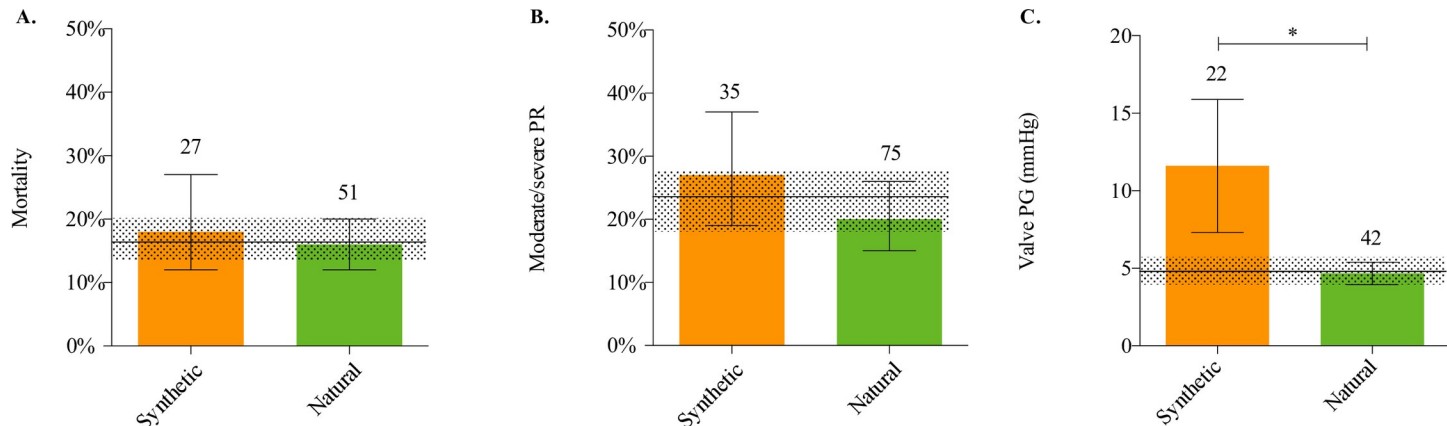

**Fig 6. Results meta-analyses natural and synthetic scaffold.** Point estimates of synthetic or natural scaffolds on mortality (**A**), reported incidence (%) of moderate or worse valve regurgitation (**B**) and peak valve pressure gradient (**C**). Overall pooled estimate (horizontal black line) and 95% CI (dotted horizontal bar). Values presented of point estimates and 95% confidence intervals. Number of comparisons on top of the bars. * *p < 0.05.*

comparison. Consequently, five out of 11 strategies could be analyzed for the valve pressure gradient. This showed a significant higher (*p = 0.01*) peak valve pressure gradient in decellularized xeno-/autografts than in bioactive/cell-seeded-on-the-fly natural scaffolds. (resp 5.25 mmHg (95% CI [4.2–6.3]) versus 3.10 mmHg 95% CI [3.7–6.3]) (Figs 6C–7C).

## Sensitivity analyses

Variation of the follow-up times of the echocardiographic assessments were visualized in box plots (S1 Fig). The mean follow-up times were qualified as equal between the synthetic and natural scaffold groups.

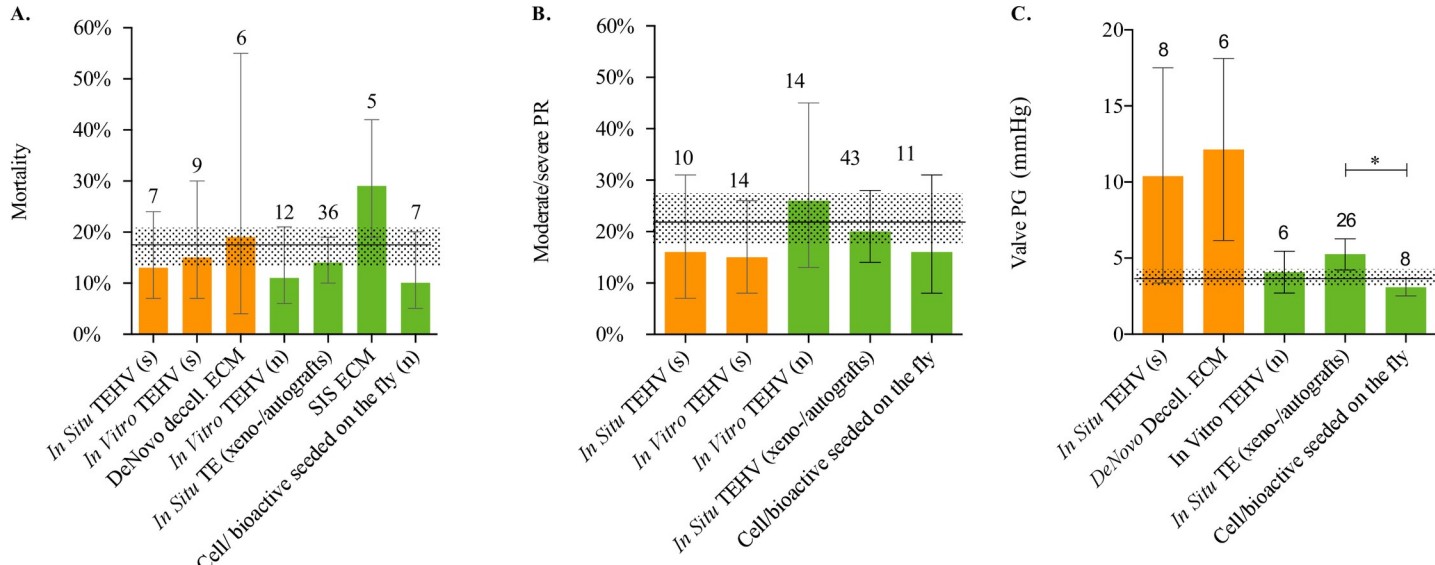

**Fig 7. Results meta-analyses of TE strategies.** Point estimates of TE strategies on mortality (**A**), reported incidence (%) of moderate or worse valve regurgitation (**B**) and mean valve pressure gradient (mmHg) (**C**). Overall pooled estimate (dotted horizontal bar). Values presented of means estimate and 95% confidence intervals and number of comparisons (top op de bars) *p < 0.05.*

Sensitivity analyses showed that, when including the studies presenting medians and ranges (and recalculate them to means and SDs), the peak pressure gradient in synthetic versus natural scaffold was no longer significant.

Sensitivity analyses on echocardiographic imaging methods (TTE, TEE, epicardial) showed no changes in the conclusions, and the data appear robust.

## Discussion

This is the first systematic review and meta-analysis summarizing all available literature on pulmonary TEHV implantations in large animal models.

This literature review clearly illustrates the large heterogeneity in study characteristics and TE strategies that have been examined. Moreover, it presents the poor reporting of essential experimental items, that hampers translation of the preclinical findings from this review to the clinical situation. Our meta-analyses showed that pressure gradients are higher in synthetic scaffolds compared to natural scaffolds. These results should however be interpreted carefully and interpreted as hypothesis generating because of the low number of included studies, high study heterogeneity, and the absence of control groups in the analysis.

The ovine model is currently by far the most used animal model for valve replacement studies. It is often chosen because of similarities with the human cardiovascular anatomy and physiology [108], as well as its ease of use [109]. Moreover, bioprosthetic valve calcification is the most frequent complication affecting patient outcomes. Because the sheep model shows rapid calcification in valve replacement [110], especially in (but not limited to) the young animal [111], therapies have generally been studied in this animal model. Still, the obtained data need to be put in perspective, as much is still unknown regarding species specific similarities and differences [112].

The mean follow-up time of the studies presented in this paper was relatively short (3.4 and 5.0 months in resp. synthetic and natural scaffold studies), and the mean number of animals per experimental group (comparison) was low (n = 3 per experimental group). Only four experimental groups (27 animals) had a follow-up time of 20–24 months [47,74,102]. While valve calcification might be evaluated in a relative short time period, TE of heart valves has the aim to create prostheses at least not inferior in durability compared to the currently available valve prostheses. In order to test extended durability of valve prostheses, it is the authors believe that preclinical *in vivo* TE studies of heart valves should preferably have long follow-up times.

An important finding in our study was that 61% of the natural scaffolds were xenografts and 39% allografts (homograft equivalent in humans). This is an important aspect in TEHV implantation because the immune response, induced by donor-recipient interaction, plays a major role in the tissue regeneration process. Xenotransplantation of natural acellular scaffolds does not necessarily induce an adverse tissue rejection response but can induce a desirable and required constructive remodeling process [113,114]. However, it can also lead to severe rejection response resulting in dramatic clinical outcomes [115]. In this regard, it is important to emphasize that animal studies using an allograft design can only be translated to a homograft-like use in patients, and must not be extrapolated to a xenograft-like situation. Moreover, in pre-clinical studies that evaluate xenograft implantation, researchers need to take into account the adaptive immuneresponse and the model specific HLA (mis)match in their chosen donor and animal model. The authors noticed that most of the studies on natural scaffold implanted as xenografts did studied the innate immuneresponse by (immune)histology (e.g., CD45, CD68, CD57) but in lesser extent evaluated the adaptive response by T-cells, B-cells (CD8, CD3, CD11b) or immunoglobulins by panel reactive antibody test.

The assessment of the reporting quality showed that there is room for improvement. Animal gender, blinding and randomization were poorly reported. In only 41% of the studies, the animal gender was clear, even though reporting the animal gender is easy and required, according to the ISO-5840 for cardiac valve prostheses [116] that can be used as basic requirements as long as no specifc ISO-standards on TEHV is available. Cardiovascular clinical trials already pay attention on biological gender difference, since they are a known important modifier [117] in relation to gender-specific inflammatory mediators involved in cardiovascular and valve diseases [118–120].

The quality assessment also revealed poor reporting of randomization and blinding methods. Randomization was reported in only five of the 46 studies applicable to use randomization [60,80,81,83,105] and when reported, the method was not always clearly described. Randomization increases the internal validity and, importantly, reduces the risk of detection bias [121]. Moreover, random allocation of animals to experimental and (if present) control groups reduces the risk of selection bias, increases the reliability of the results, and is a requirement for an appropriate experimental design when interventions are being compared [122]. Measures to ensure blinding of the investigators and other personnel is often poorly reported in animal studies [123,124]. We have identified only two studies [71,94] that reported blinding of the assessors on valve functionality. Blinding is especially important when it comes to qualitative outcome assessment, particularly if there is a subjective element in the outcome like echocardiographic evaluation of the valve or reading histological slides, both important in TEHV studies. We understand that blinding is not always possible in the surgical procedure of valve implantations. However, description of blinding (or the reason to not blind), always is.

While evaluating the presence of a control group, it appeared that no standard type of control group is used in TEHV research. Standardization of a control valve would improve the comparability between studies. However, the use of control animals is expensive and raises ethical concerns regarding the number of animals. Moreover, depending on the research question and stage of the study, the necessity and type of control valve differs.

Furthermore, it should be taken into account that valve degeneration is faster in children than in adults. The choice of the control animal must match here.

The use of a (clinical accepted) biological prosthesis as comparative control, is one (and probably the most important) option [125]. Indeed, bioprostheses can serve as control for valve function and durability. However, they lack information regarding formation/engineering of valve tissue, which is specific for TEHV. In our opinion, it is time to gain consensus on these vital quality items that will reduce risks of bias and improve interpretation and translatability of the studies. Many initiatives have been developed to support researchers and journal editors to improve the quality of animal studies [126]. For example, by pre-registration of planned animal studies (https://preclinicaltrials.eu) or using the ARRIVE reporting checklist [127,128].

We performed a meta-analysis on mortality and valve functionality. This was not to obtain a precise point estimate, but rather to get an impression whether or not the various scaffolds used in the preclinical setting may differ in functionality. Until now, validated data of (expected) mortality in pulmonary valve implantations in large animals was lacking. Our study shows an overall pooled estimated mortality of 17%. The meta-analysis showed no significant differences in mortality between the groups. The incidence of (unplanned) mortality are helpful in preparation and calculation of the numbers of animals to include in future studies. Moreover, the cause of mortality (see supplement) was mainly either operation related or due to endocarditis. This highlights the importance of trained personnel and sterility.

The meta-analysis showed that synthetic scaffolds had a higher mean pressure gradient compared to natural scaffolds. This difference is not unexpected, since the natural valve ECM

micro-architecture is very important for leaflet flexibility and mechanical strength [129–131]. The natural micro-architecture has not yet been replicated in synthetic scaffolds. Still, the estimated pressure gradient in both groups is low and in the range of normal human pulmonary valve mean pressure gradients (< 25 mmHg in rest or <30 mmHg in exercise). Also, during sensitivity analyses, peak pressure gradient in synthetic valve remained higher compared to natural scaffolds, however, the difference was no longer statistically significant.

Research on TEHV exist nearly 25 years and studies enter the clinical stage [132]. In our opinion, it is time to have a discussion with a group of experts, and strive towards standardization of preclinical large animal studies (e.g., control group, follow-up time) and animal-free (or friendly) alternatives. With this review, we highlight the importance of good reporting in animal studies. Adequate reporting and standardization will greatly enhance the possibilities for meta-analysis and support safe translation to the clinic.

## Limitations and future implications

Several limitations appear in this systematic review and meta-analysis. First, the studies are very heterogeneous in design. Heterogeneity in animal studies can be expected, more so than a typical clinical trial because of the often-exploratory approach [133]. To account for this heterogeneity, we used a random effects model, and explored the suggested causes for study heterogeneity by means of subgroup analyses. Evaluation of this heterogeneity is one of the added values of meta-analysis of animal studies and might help to design future animal studies and subsequent clinical trials. However, successfully translating findings to the clinical arena largely depends upon an understanding of the sources of heterogeneity, and their impact on effect size. We made a start by presenting an overview of these (heterogeneous) study characteristics. Still, future (sensitivity) analyses on animal species, scaffold topographical characteristics or cell type could give insight in the relation of each of the items on the outcomes. Second, due to a low number of comparisons in the meta-analysis, the estimated summary effect may be imprecise. As a final point, because no standard control groups in the studies were used, we could not calculate an effect estimate between groups, only an estimate of the individual experimental group. Because of these mentioned items, interpretation of the outcome should be taken with caution.

## Conclusion

This systematic review summarizes all available literature on pulmonary TEHV implantation in large animals. We showed that there is substantial heterogeneity in study designs and TE strategies between the included studies. Moreover, it shows that the methodological quality and quality of reporting can be improved by providing more detailed description of animal characteristics and blinding and randomization methods.

The meta-analysis revealed that the transvalvular pressure gradient was significant higher in synthetic scaffolds. However, these results should be interpreted with caution due to substantial heterogeneity in the design of the included studies, and the relatively small number of included studies. To move the TEHV field forward and enable reliable comparisons, it is essential to define standardized methods and ways of reporting. This would greatly enhance the value of individual large animal study.

## Supporting information

**S1 Checklist. PRISMA 2009 checklist.**
(PDF)

**S1 Fig. Follow-up time of each comparison in the meta-analyses.** Follow-up time of each comparison in the meta-analyses showed equal variation in the natural and synthetic scaffolds in valve regurgitation **(A)** and pressure **(B)** gradient analyses.
(DOCX)

**S1 Table. Syntax OVID Medline and Embase database.**
(DOCX)

**S2 Table.** A. Questionnaire of the quality assessment. B. Questions on the control groups.
(DOCX)

**S3 Table. Reported anticoagulation therapies.**
(DOCX)

**S4 Table.** A. Study characteristics of the pre-clinical studies on synthetic scaffolds. The applied strategy number correlates with the numbers in Fig 4. NR; Not reported. Ns: Not specified. PV; Pulmonary valve. AV: Aorta valve. PVR: Pulmonary valve replacement. *AV*: *Aorta valve. BU; Bis-urea ECM; Extracellular matrix P(L,DL)LA; Poly(L-lactide-co-D,L-lactide). P4HB; Poly-4-hydroxybutyrate PC; Poly-carbonate PCL; Polycaprolactone PCUU; Poly-carbonate urethene urea PDO; Poly(1,4-dioxan-2-one) PGA; Polyglycolic Acid. PHO; Polyhydroxyoctanoate PV; Pulmonary valve. SIS; Small intestinal submucosa TE; Tissue engineering TEHV; Tissue engineered heart valve. UPy; 2-ureido-4[1H]-pyrimidinone.* B. Study characteristics of the pre-clinical studies on natural scaffolds. *AV; Aorta valve*, F; *Female, M; Male, Ns: Not specified, PV; Pulmonary valve, PVR: Pulmonary valve replacement, RVOT; Right ventricle outflow trac,. SIS-ECM; Small intestinal submucosa-extra cellular matrix, TE; Tissue engineering.*
(DOCX)

**S5 Table.** A. Tissue engineering strategies of the studies on synthetic scaffolds. *BMD; Bone marrow derived, CCN-1; Cysteine-rich angiogenic protein 61. EDTA; Ethylenediamine tetraacetic acid, EPC; Endothelial progenitor cell, FB; Fibroblasts, MNCs; Mononuclear cells, N/A; Not applicable, PD; Pulse duplicator, SMCs; Smooth muscle cells.* B. Tissue engineering strategies of the studies on natural scaffolds. *BMD; Bone marrow derived. CCN-1; Cysteine-rich angiogenic protein 61. EDTA; Ethylenediamine tetraacetic acid. EPC Endothelial progenitor cell; FB; Fibroblasts. MNCs; Mononuclear cells. N/A; Not applicable. PD; Pulse duplicator. SMCs; Smooth muscle cells.*
(DOCX)

**S6 Table. Results of quality assessment of preclinical studies.** Results showing that study design such as inclusion of proper control groups, randomization and blinding, reporting of key items on animal characteristic such as gender and strain and needs to be improved in pre-clinical studies on TEHVs.
(DOCX)

**S7 Table. Results of reported mortality of the synthetic and natural scaffolds showing that the most often operation related cause of (unplanned) death was reported.**
(DOCX)

## Acknowledgments

The authors would like to thank Jacqueline Limpens and Alice Tillema for conducting the search syntax and performing the search in the medical databases, and Annemijn Vis and Andras Durko for assistance in the quality assessment.

## Author Contributions

**Conceptualization:** M. Uiterwijk, J. Kluin.

**Data curation:** M. Uiterwijk, R. van Vliet, I. J. de Brouwer.

**Formal analysis:** M. Uiterwijk, C. R. Hooijmans.

**Funding acquisition:** M. Uiterwijk.

**Investigation:** M. Uiterwijk, D. C. van der Valk, R. van Vliet, I. J. de Brouwer.

**Methodology:** M. Uiterwijk, R. van Vliet, I. J. de Brouwer, C. R. Hooijmans.

**Project administration:** M. Uiterwijk, R. van Vliet, I. J. de Brouwer.

**Resources:** M. Uiterwijk, R. van Vliet, I. J. de Brouwer, C. R. Hooijmans.

**Software:** M. Uiterwijk, C. R. Hooijmans.

**Supervision:** M. Uiterwijk, C. R. Hooijmans, J. Kluin.

**Validation:** M. Uiterwijk, D. C. van der Valk, C. R. Hooijmans.

**Visualization:** M. Uiterwijk, D. C. van der Valk.

**Writing – original draft:** M. Uiterwijk.

**Writing – review & editing:** M. Uiterwijk, D. C. van der Valk, R. van Vliet, I. J. de Brouwer, C. R. Hooijmans, J. Kluin.

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
