## [Decision Letter · Decision Letter 0]

5 Jun 2021

PONE-D-21-04019

PULMONARY VALVE TISSUE ENGINEERING STRATEGIES IN LARGE ANIMAL MODELS

Systematic Review, Quality Assessment & Meta-Analysis

PLOS ONE

Dear Dr. Uiterwijk,

Thank you for submitting your manuscript to PLOS ONE. After careful consideration, we feel that it has merit but does not fully meet PLOS ONE’s publication criteria as it currently stands. Therefore, we invite you to submit a revised version of the manuscript that addresses the points raised during the review process.

We look forward to receiving your revised manuscript.

Kind regards,

Panayiotis Maghsoudlou

Academic Editor

PLOS ONE

Journal Requirements:

This work was funded by ZonMW, more knowledge with fewer animals (Projectnumber 114024134, 40-42600-98-433). C.R. Hooijmans is employee of SYRCLE. M. Uiterwijk is support from the Netherlands Cardiovascular Research Initiative (CVON 2012-01): The Dutch Heart Foundation, Dutch Federation of University Medical

451 Centers, the Netherlands Organization for Health Research and Development and the Royal Netherlands Academy of Sciences. D.C. van der Valk is supported by the Gravitation Program “Materials Driven Regeneration”, funded by the Netherlands Organization for Scientific Research (024.003.013).

This work was funded by ZonMW, more knowledge with fewer animals (Projectnumber 114024134, 40-42600-98-433). C.R. Hooijmans is employee of SYRCLE. M. Uiterwijk is support from the Netherlands Cardiovascular Research Initiative (CVON 2012-01): The Dutch Heart Foundation, Dutch Federation of University Medical Centers, the Netherlands Organization for Health Research and Development and the Royal Netherlands Academy of Sciences. D.C. van der Valk is supported by the Gravitation Program “Materials Driven Regeneration”, funded by the Netherlands Organization for Scientific Research (024.003.013).

Reviewers' comments:

Reviewer's Responses to Questions

**Comments to the Author**

1. Is the manuscript technically sound, and do the data support the conclusions?

Reviewer #1: Yes

Reviewer #2: Yes

2. Has the statistical analysis been performed appropriately and rigorously? 

Reviewer #1: Yes

Reviewer #2: Yes

3. Have the authors made all data underlying the findings in their manuscript fully available?

Reviewer #1: Yes

Reviewer #2: Yes

4. Is the manuscript presented in an intelligible fashion and written in standard English?

Reviewer #1: Yes

Reviewer #2: Yes

5. Review Comments to the Author

Reviewer #1: The authors provided a systematic review, quality assessment, and meta-analysis regarding publications on tissue engineering heart valve strategies in large animal models. This review was presented in a clear, concise, and informative way that highlights changes necessary in this area of research to provide more consistent, clear, and quality translational and preclinical work. The reviewer recommends publication of this work that will likely guide better large animal study design but has a few minor comments to improve clarity of the work presented. Overall, the study design was very transparent.

Minor comments:

1. Please read through the entire text for punctuation errors (primarily with missing punctuation), correct use of plural, and tense agreement.

a. Examples: line 39 “publications”, line 140 “questions”, lines 194,195 consistency with number punctuation, 229 “subcutaneously”, line 73, 364, 376

2. Figure 2A- caption states “shifted from mainly natural-based scaffold design before 2015, to synthetic designs after 2015,” but this wording is not clear. The majority of the studies still use natural scaffold (>50%), but the proportion is changing over time. Please be more precise with the wording.

3. Figure 3- consider specifying what (n) and (s) mean explicitly in the figure caption.

4. Figure 4- Please reword this caption as this was extremely difficult to understand.

5. Figure 6 and 7- consider adding the figure title as the y-axis label for clarity.

6. Discussion:

a. Of the studies where gender was reported, was the castration/intact status of the animals included? This information could be of interest in interpreting the success and/or calcification of the implanted valves.

b. From the collected data, would it be possible to determine calcification status from the natural vs. synthetic valve implants in these animal models?

Reviewer #2: Overall an excellent contribution by Uiterwijk et al and discussion covers many important points.

Few topic if Authors can cover a bit more in details.

1: In discussion author cover about xenotransplant and how generally it can tolerate in animal model but hasnt worked in clinic. Question is, does any of the study actually study the immune response as in evaluate both innate and adaptive immuneresponse, monitor IgG against graft, development of HLA antigen against implanted tissue. It needs to be highlighted that poorly derived conclusions with xeno animal studies can lead to major adverse events in clinic.

2: Great discussion on need for control. It could be expanded with more discussion if valve is for Pediatric vs. adult population and how animal age can impact outcome.

3: ISO 5840 2021 has very clear requirement for control valve in addition to other very specific in vivo study requirement. Authors do bring the standard once, which is great. Perhaps a paragraph towards the end with bit more context from new ISO standard (issued Jan 2021) can help further standardize future animal studies.

Overall, great work and really appreciate authors making an effort to put this together.

6. PLOS authors have the option to publish the peer review history of their article (what does this mean?). If published, this will include your full peer review and any attached files.

Reviewer #1: No

Reviewer #2: **Yes: **Dr. Zeeshan Syedain

---

## [Author Response · Author response to Decision Letter 0]

29 Jul 2021

Reviewer #1: The authors provided a systematic review, quality assessment, and meta-analysis regarding publications on tissue engineering heart valve strategies in large animal models. This review was presented in a clear, concise, and informative way that highlights change necessary in this area of research to provide more consistent, clear, and quality translational and preclinical work. The reviewer recommends publication of this work that will likely guide better large animal study design but has a few minor comments to improve clarity of the work presented. Overall, the study design was very transparent.

Minor comments:

1. Please read through the entire text for punctuation errors (primarily with missing punctuation), correct use of plural, and tense agreement.

a. Examples: line 39 “publications”, line 140 “questions”, lines 194,195 consistency with number punctuation, 229 “subcutaneously”, line 73, 364, 376

Response: We read the manuscript carefully and made the necessary corrections. 

2. Figure 2A- caption states “shifted from mainly natural-based scaffold design before 2015, to synthetic designs after 2015,” but this wording is not clear. The majority of the studies still use natural scaffold (>50%), but the proportion is changing over time. Please be more precise with the wording.

Response: 

We changed the caption of Figure 2A from: 

“Percentage of publications of studies on TEHV scaffolds shifted from mainly natural-based scaffold designs before 2015, to synthetic scaffolds designs after 2015”

Into 

“The proportion of publications on synthetic TEHV scaffolds increased overtime”

3. Figure 3- consider specifying what (n) and (s) mean explicitly in the figure caption.

Response: 

We changed the legend item in figure 3A (details can be find in the uploaded file :respons to the reviewer for the figure)

4. Figure 4- Please reword this caption as this was extremely difficult to understand.

Response: 

We reword the caption in figure 4 as follow.

From: Figure 4. Overview of strategies in tissue engineering of heart valves. Start with the scaffold materials (synthetic or natural), subsequent the pathways of preparation techniques prior to implantation. Finally present 11 strategies were identified, n= numbers of valves represented in each group. TE; Tissue engineering. SIS-ECM; Small intestinal submucosa extra cellular matrix. *Decellularized. Xeno-/autografts

Into: Figure 4. Overview of strategies in tissue engineering of heart valves. Based on all experimental groups found in literature, 11 different TE strategies were identified. Five strategies used a natural scaffold (green; n) and six strategies a synthetic (orange; s) scaffold. TE; Tissue engineering. SIS-ECM; Small intestinal submucosa extra cellular matrix. *Decellularized. Xeno-/autografts

5. Figure 6 and 7- consider adding the figure title as the y-axis label for clarity.

Response: 

We changed the figure and replaced the title to the y-axis.

6. Discussion:

a. Of the studies where gender was reported, was the castration/intact status of the animals included? This information could be of interest in interpreting the success and/or calcification of the implanted valves.

Response: 

This is an interesting question. The castration status of the animals was not reported in any of the studies as far as the authors remember. Although some authors mentioned “fertile” status of the animals. Although, not further specified. This is an important topic because their fertility is depending on age (Age at puberty (ewe): 6 - 9 months) amongst other factors like season and the strain. 

However, we did not pre-defined and did not systematic analyzed this question and so did not double checked this data. Therefor we do not want to speculate on this topic in the manuscript. Especially because the effect of castration and the hormonal effects of the animal in relation to the regeneration process is although interesting also yet unknown and complicated and reaching out of the topic of the subject of this manuscript. 

b. From the collected data, would it be possible to determine calcification status from the natural vs. synthetic valve implants in these animal models?

Response: 

This is an excellent suggestion and we are happily to announce that in the past months one of the Co-authors (D. van der Valk) used the literature data set to collect all outcomes regarding calcification and subsequently conducted a new meta-analysis. The manuscript will be submitted shortly after the current paper is accepted.

 

Reviewer #2: Overall an excellent contribution by Uiterwijk et al and discussion covers many important points.

Few topic if Authors can cover a bit more in details.

Response: 

Thank you for these kind words.

1: In discussion author cover about xenotransplant and how generally it can tolerate in animal model but hasn’t worked in clinic. Question is, does any of the study actually study the immune response as in evaluate both innate and adaptive immuneresponse, monitor IgG against graft, development of HLA antigen against implanted tissue. It needs to be highlighted that poorly derived conclusions with xeno animal studies can lead to major adverse events in clinic.

Response: 

The authors do agree with the reviewer that care should be taken on the translatability of results from animal derived studies on xenograft implantation. Special care needs to be taken regarding the innate but also humoral response. As the innate immuneresponse is often evaluated by (immune)histology (e.g., CD45, CD68, CD57), in lesser extent the adaptive response of T-cells and B-cells (CD8, CD3, CD11b). 

The first author (M.U) scanned the publication on natural valves implanted as xenograft. This elucidated that, 28 publications of natural scaffolds implanted as xenograft described an evaluation of the innate immuneresponse and 8 studies (additionally) evaluated the adaptive immune response. Although this subject is not our expertise and not predefined in our study protocol, the authors do agree that some more attention can be made on this subject in the discussion, although with some caution.

Therefor we added the following sentence:

“Moreover, in pre-clinical studies that evaluate xenograft implantation, researchers need to take into account the adaptive immuneresponse and the model specific HLA (mis)match in their chosen donor and animal model. The authors noticed that most of the studies on natural scaffold implanted as xenografts did studied the innate immuneresponse by (immune)histology (e.g., CD45, CD68, CD57) but in lesser extent evaluated the adaptive response by T-cells, B-cells (CD8, CD3, CD11b) or immunoglobulins by panel reactive antibody test.” (Line 462-467)

2: Great discussion on need for control. It could be expanded with more discussion if valve is for Pediatric vs. adult population and how animal age can impact outcome.

Response: 

Thank you for this suggestion. The authors do agree and added the mentioned topics for the readers when considering the use of a control valve: 

“Furthermore, it should be taken into account that valve degeneration is faster in young children than in adults. The choice of the control animal must match here.” (Line 495-496)

3: ISO 5840 2021 has very clear requirement for control valve in addition to other very specific in vivo study requirement. Authors do bring the standard once, which is great. Perhaps a paragraph towards the end with bit more context from new ISO standard (issued Jan 2021) can help further standardize future animal studies.

Response: 

Thank you for this suggestion. We replaced the reference to the new ISO standards (Line 356). Although the ISO standards are not developed for tissue engineered heart valves, in our opinion the standards can be can be the basis for TEHV and can be further customized with specific requirements, e.g., regarding analyzes of the recipient immune response. We added this in line 417 the following sentence:

“that can be used as basic requirements as long as no specific ISO-standard on TEHV is available” (Line 471)

Overall, great work and really appreciate authors making an effort to put this together.

---

## [Editor Report · Decision Letter 1]

17 Sep 2021

Pulmonary valve tissue engineering strategies in large animal models

Systematic review, quality assessment & meta-analysis

PONE-D-21-04019R1

Dear Dr. Uiterwijk,

We’re pleased to inform you that your manuscript has been judged scientifically suitable for publication and will be formally accepted for publication once it meets all outstanding technical requirements.

Kind regards,

Panayiotis Maghsoudlou

Academic Editor

PLOS ONE

---

## [Editor Report · Acceptance letter]

22 Sep 2021

PONE-D-21-04019R1 

Pulmonary valve tissue engineering strategies in large animal models
Systematic review, quality assessment & meta-analysis 

Dear Dr. Uiterwijk:

I'm pleased to inform you that your manuscript has been deemed suitable for publication in PLOS ONE. Congratulations! Your manuscript is now with our production department. 

Kind regards, 

on behalf of

Dr. Panayiotis Maghsoudlou 

Academic Editor

PLOS ONE